cognition, behaviour

GPS tracking, homing pigeons, spatial cognition, cultural transmission, collective learning, memory duration

**Author for correspondence:**
Julien Collet
e-mail: pro@colletj.fr

# Pigeons retain partial memories of homing paths years after learning them individually, collectively or culturally

Julien Collet[1], Takao Sasaki[2] and Dora Biro[1,3]

[1]Oxford Navigation Group, Department of Zoology, University of Oxford, 11A Mansfield Road, Oxford OX1 3SZ, UK
[2]Odum School of Ecology, University of Georgia, Athens, GA, USA
[3]Department of Brain and Cognitive Sciences, University of Rochester, Rochester, NY, USA

JC, 0000-0002-3285-5535; TS, 0000-0001-7923-9855; DB, 0000-0002-3408-6274

Memory of past experience is central to many animal decisions, but how long specific memories can influence behaviour is poorly understood. Few studies have reported memories retrieved after several years in non-human animals, especially for spatial tasks, and whether the social context during learning could affect long-term memory retention. We investigated homing pigeons' spatial memory by GPS-recording their homing paths from a site 9 km from their loft. We compared solo flights of naive pigeons with those of pigeons that had last homed from this site 3–4 years earlier, having learnt a homing route either alone (individual learning), together with a naive partner (collective learning) or within cultural transmission chains (cultural learning). We used as a control a second release site unfamiliar to all pigeons. Pigeons from all learning treatments outperformed naive birds at the familiar (but not the unfamiliar) site, but the idiosyncratic routes they formerly used several years before were now partially forgotten. Our results show that non-human animals can use their memory to solve a spatial task years after they last performed it, irrespective of the social context during learning. They also suggest that without reinforcement, landmarks and culturally acquired 'route traditions' are gradually forgotten.

## 1. Introduction

Perception of and interactions with the environment generate individual experiences that can inform subsequent behavioural decisions in all animals [1]. How long after an experience an animal can still retrieve and use the associated memory for its decisions is a crucial parameter in predicting behaviour and understanding its consequences. At the upper extreme, reports of memory lasting several years or longer (e.g. [2]) are still uncommon in non-human animals (see the electronic supplementary material, table S1); moreover, they usually consist of observations from a few individuals and are rarely comparative in nature (but see [3,4]). As a result, factors affecting whether and which memories can persist long term (e.g. over several years) are rather poorly understood.

Memory relies on changes in the organization and strength of synaptic connections, but these changes are not necessarily permanent or permanently accessible for retrieval [5,6]. For instance, successive experiences accumulating over time can compete and interfere with each other for access to long-term storage and/or retrieval [7,8]. Such competition effects could result in selectivity in what is remembered or retrieved (e.g. primacy or recency effects if, respectively, early or late experiences are more likely to be remembered [9,10]). Memory retention may also depend on the tasks involved and the conditions experienced during learning [11–13], as they may not necessarily involve the same neural pathways [14,15] and therefore the same neuronal stability over time.

Hence it is important to empirically study memory retention for a range of tasks and contexts.

Spatial memory, in particular, can affect a wide range of ecological processes, from global migration patterns to population dynamics [16]. It is unclear how common very long-term spatial memories are in animals. There are anecdotal reports from the wild, for instance, of two elephant clans returning to distant waterholes visited only during extreme drought events [17]. However, in the wild, it is generally challenging to robustly test the use of memory [16] and to discard competing hypotheses (e.g. the effects of untracked conspecifics and/or environmental cues, unmeasured reinforcement opportunities, genetically inherited navigation programmes, etc.). Still, many fishes, turtles and birds are known to return to their natal sites after several years of dispersal ('philopatry': [18,19]), presumably involving the use of memory. The current main hypothesis is that philopatry occurs through magnetic or olfactory gradient-climbing (a rather simple form of spatial cognition) and imprinting (i.e. using a critical memory window at birth) on the magnetic or olfactory signature of the natal location [18,19]. It is unclear whether the mechanisms involved in philopatry could extend to non-natal areas experienced later in life by animals, and/or compare in retention with mechanisms of spatial memory other than gradient-climbing. In particular, many animals can rely on remembering landmarks and routes (e.g. homing pigeons: [20,21]), a cognitive process involving very specific areas of the brain [21–24]. Beyond philopatric animals [18,19], we did not find controlled experiments reporting spatial memory persisting over multiple years without reinforcement (see the electronic supplementary material, table S1). This gap in reports compared to other types of memory tasks (electronic supplementary material, table S1) might be because the specific neural substrates often involved in spatial memories [22–24] can undergo important age-related [25] and/or seasonal neural changes [6], and might be less stable over years than other types of memories. Alternatively, it could simply be the result of logistical challenges of testing subjects on a spatial task several years after learning.

Moreover, no studies have examined how long-term memory retention is affected by the social context in which learning occurs (electronic supplementary material, table S1, see also [26]). Within a group, individuals can learn from the behaviour of other, more experienced members of the group ('social learning', [27,28]), which may sometimes involve specific neural pathways [26] and may therefore affect how long these memories can persist. It has also been observed that groups of initially all-naive individuals can improve their collective efficiency when solving the same task repeatedly ('collective learning', [29–31]), but in these cases, it can be difficult to determine how memory is distributed within the group (i.e. who learnt what [29]). Moreover, the social interactions involved in maintaining group cohesion could affect the individual learning process itself [32]. In particular, within a group acting collectively, individual members may each experience a weaker association between their own contribution and the resulting costs or benefits, compared to individual learners [33,34], and it is known that weaker associations between cues and rewards can affect long-term storage and/or retrieval of memories [35]. In theory, the social context during learning might therefore affect individual memory retention in various ways.

Here, we tested whether domestic pigeons (*Columba livia*) could remember how to home from a specific release site 8.6 km from their loft, a task they had learnt several years previously. Numerous experiments suggest that in pigeons such learning involves memorization of a chain of landmarks, i.e. a route (reviewed extensively in [20]). These routes develop gradually, stabilize without reaching maximum efficiency (i.e. are not the straight-line path between the release site and home), and are idiosyncratic (i.e. different individuals develop different routes, which they recapitulate faithfully) [20]. However, to our knowledge their retention has only ever been formally tested over a maximum of a few days or weeks ([20], see also [36]). During an experiment conducted in 2016 [37], pigeons had developed and learnt their own idiosyncratic routes from a release site by flying either alone (individual learning), along with an initially equally naive partner (collective learning), or along with a more experienced partner that had already learnt a route before (cultural learning). Here, we released these same subjects at the same site, 3–4 years later (in 2019 and 2020), in a single solo flight. Importantly, these birds were not given any experience with the release site in the intervening years. We compared their homing efficiency and the retention of their routes to that observed during their last flights in 2016, as a function of the social context that they had experienced during learning. We also compared the homing efficiency (i.e. probability to successfully home and homing flight distance) of these experienced birds to that of naive birds, both at the release site used in 2016 and at a second, control site where none of our subjects had previously flown from.

## 2. Material and methods

### (a) Study site and system

Our study used pigeons from a captive population bred at the John Krebs Field Station at Wytham, near Oxford, UK ('home lofts', 51.7828° N, 1.3174° W; see the electronic supplementary material for detailed husbandry facilities and procedures). This population of pigeons has been regularly used for homing experiments over the past 30 years, where pigeons are captured, driven away from their home loft (at typical distances of *ca* 10 km) and released alone or in groups [20], homing paths being now recorded by small GPS units attached to each individual.

In each experimental year, all pigeons selected as subjects followed a two-month standard pre-training procedure prior to the start of the experiments [20,37] where they were released from sites situated 2 km from their loft in all four cardinal directions. Each pigeon received four flock releases followed by four solo releases from each training site. This pre-training was intended to both improve pigeons' body condition for subsequent longer-distance releases, and to familiarize them with the general experimental procedure and with the area surrounding their loft. All releases, training and experimental, were undertaken while the sun's disc was visible and wind speeds were below 10 ms$^{-1}$.

### (b) Learning phase (in 2016) and social context categories

In 2016, pigeons were released from Greenhill Farm (51.8563° N, 1.2843° W, direction and distance to loft: 197°, 8.6 km). The purpose of this original study was to compare the homing behaviour along successive releases between three different experimental treatments: a solo-control (pigeons released alone 60 times), a pair-control (pairs of initially naive pigeons released always together, 60 times) and a cultural chain of transmission. For the cultural chain treatment, in each 'chain replicate' a single

**Table 1.** Data structure and sample sizes for the initial learning conditions in 2016 [37] and the test phase and present analyses in 2019–2020.

| learning category in this study | experimental treatment in 2016 | sample size at Greenhill Farm in 2016 [37] | | | sample size at Greenhill Farm (GH) in 2019–2020 | | | | sample size at Horspath (HP) in 2020 | | |
|---|---|---|---|---|---|---|---|---|---|---|---|
| | | n replicates in 2016 | n individuals in 2016 | n releases per individual | n individuals released at GH in 2019 | n individuals released at GH in 2020 | n individuals back home (2019–20) | n usable tracks in 2019–20 at GH | n individuals released at HP in 2020 | n individuals back home | n usable tracks in 2020 at HP |
| individual learning | solo control | 9 | 9 | 60 | 2 | 1 | 2 | 2 | 1 | 1 | 1 |
| | cultural chain 1st generation | 8 | 8 | 24 | 3 | 2 | 5 | 5 | 2 | 2 | 2 |
| collective learning | pair control | 6 | 12 | 60 | 1 | 3 | 4 | 4 | 3 | 3 | 2 |
| cultural learning | cultural chain 2nd–5th generations | 8 | 32 | 24 (gen. 2–4) 12 (gen. 5) | 2 | 18 | 20 | 15 | 18 | 18 | 13 |
| naive | not applicable | 0 | 0 | not applicable | 44 | 14 | 40 | 35 | 10 | 9 | 6 |

pigeon was first released 12 times alone (1st generation), then released 12 more times with an initially inexperienced partner (2nd generation), then for the next 12 flights (3rd generation), the bird from the 1st generation was removed and a third, initially inexperienced individual paired with the 2nd-generation bird. This chain continued by replacing the 'oldest' member of the pair by an inexperienced one every thirteenth release, up to 'generation 5' (and thus a total of 60 flights per chain). The outcome of this experiment, along with methodological details can be found in [37] and the sample sizes are summarized in table 1. Note that when released in pairs, virtually all pigeons flew cohesively together in 2016 (never flying more than 150 m apart, [37,38]); those that did not were not used for the present experiment.

Here, we consider this 2016 experiment as our 'learning phase'. Depending on their experimental role in 2016, pigeons experienced different social contexts during learning: some developed their routes alone, others with a stable partner, and yet others experienced changes in partnership after 12 flights; among pigeons flying in pairs, some could initially learn alongside a more experienced partner, some developed their experience jointly, and some had to fly with a new, inexperienced partner after they had developed a stable route [20,31,37]. Individual pigeons also differed in the total number of flights undertaken in 2016 (12, 24 or 60; table 1). Not all pigeons used in 2016 were still available in 2019–2020, and our reduced sample size forced us to simplify these various social contexts (and various total numbers of flights) into three main categories for statistical analyses (table 1).

We categorized as 'individual learning' pigeons from the solo control treatment and 1st-generation pigeons from the cultural chain treatment (i.e. all pigeons that performed their first 12 flights alone, the phase where homing routes undergo most variation before becoming more stable [20,31,37]). We categorized as 'collective learning' pigeons from the pair-control treatment (i.e. pigeons that always flew in pairs with always the same, equally experienced partner). We categorized as 'cultural learning' pigeons from generations 2 to 5 from the cultural chain treatment (i.e. pigeons that performed their first 12 flights alongside a more experienced partner).

Our categorization was decided upon after exploring various options. In the electronic supplementary material, figures S2–S4, we show data and results for each generation of birds separately (i.e. maximum number of categories), to graphically show that our choice of categorization does not affect our conclusions. In particular, in support of our claim that the exact categorization matters little, we found no indication that after several years birds were more likely to remember either their earlier or their later homing paths, regardless of the learning treatment (electronic supplementary material, figure S4; i.e. no primacy/recency-like effects).

## (c) Memory retention tests (in 2019–2020)

In 2019–2020, we released at the same site (Greenhill Farm) pigeons that had been part of the 2016 experiment ('experienced' birds) and pigeons that had never been released at Greenhill Farm ('naive' birds). In 2019, the experimental plan (beyond the scope of this manuscript) was not focused on testing very long-term memory, and we rather prioritized the recruitment of naive birds (see table 1 for sample sizes). In 2020, we more thoroughly tested the long-term homing memory of experienced pigeons, by recruiting all birds available for experiments, not already released in 2019, and that had either been part of the 2016 experiment ('experienced' birds), or had never been released at Greenhill Farm ('naive' birds; table 1 for sample sizes). Finally, to check that our results were not a consequence of age or general experience at homing (control naive birds being on average younger than experienced birds), we released

all the birds that had returned from the Greenhill Farm test release (see table 1 for sample sizes) at a second, control release site that all birds were naive with (Horspath: 51.7366° N, 1.1853° W; electronic supplementary material, figure S1). Horspath is at a similar distance from the home lofts (10.4 km) as Greenhill Farm but in roughly the opposite direction (direction from lofts: 301°).

## (d) GPS deployments and settings

Pigeons carried on their back a small GPS unit weighing 15 g (BT-Q1300ST, Qstarz, Taiwan; *ca* 2–5% of the pigeons' body mass), attached either through hand-made adjustable backpacks (2016 and 2019) or through velcro strips glued to their back feathers (2020). After homing, birds were captured again to retrieve the GPS devices. GPS devices were set to record at 5 Hz in 2016 and 2019. These recording frequencies rapidly saturated onboard device memory so that many flight tracks during early releases, especially of naive birds who can take many hours or even days to return home, were incomplete (stopping after 4–6 h). To record a larger proportion of early flights, in 2020, we set the GPS to record at 1 Hz frequency (at which point battery life became the limit, lasting approx.10 h).

## (e) Data curation

At the end of each season (2019 and 2020), we identified which individuals never came back to the loft (losses), for which no data is available (see table 1 for sample sizes). For birds that did return, we checked if GPS records were complete (i.e. ending less than 200 m from the loft), which was taken into account in subsequent analyses (see below). We also checked that solo-released pigeons did not accidentally join with other individuals released before them: we considered that flights with greater than 75% of locations greater than 150 m from any other simultaneous pigeon GPS location could be considered solo flights [37,38], and we discarded non-solo flights from subsequent analyses (see table 1 for resulting sample sizes).

For analyses, all GPS tracks from 2016 and 2019 were down-sampled to 1 Hz to match the lower GPS recording frequency used in 2020. We used a universal transverse mercator spatial projection of longitude and latitude for all spatial analyses. Pigeons usually perform initial circling over the release point before choosing a homing direction, and may circle again once they reach the home lofts, prior to landing. To remove the effect that such non-goal-directed sections of the track have on our calculations of homing efficiency, similar to previous studies [37] we identified the first point beyond 200 m of the release point and the first point within 50 m of the loft and only kept for analyses the flight path in between.

## (f) Analyses of homing efficiency

Homing efficiency was measured by a straightness index, calculated as the straight-line distance between the release site and the home lofts divided by the cumulative distance flown by the bird as recorded at 1 Hz. This index is between 0 and 1, and approaches 1 as the homing path approaches the straight line to home (considered maximum efficiency). As in previous studies, we retained incomplete tracks by assuming a straight line to home from the last recorded GPS location. This inevitably over-estimates homing efficiency of incomplete tracks, but for our conclusions this is conservative. Indeed, incomplete tracks usually wander far off the straight line home (low efficiency), and incomplete tracks occurred in larger proportions within naive than experienced birds at the test release site (see Results), so the over-estimation of efficiency should be more pronounced in naive birds than in experienced birds, reducing (rather than inflating) our chances of detecting a difference if there is one.

We used Gaussian linear models taking homing efficiency as the response variable. Owing to the complexity of the design, not all explanatory variables of interest (naive versus experienced; 2019 versus 2020; 2019–2020 versus 2016; for all experienced birds pooled together or for each of the learning conditions) could be included simultaneously in a single model, so we ran several separate models, each described in the Results section where appropriate. Separate models were used for each release site. We used a Gaussian structure for all these models even though straightness is bounded between 0 and 1 because averages were sufficiently far from the edges relative to spread; all predicted values of our statistical models fell between 0 and 1.

We also used $\chi^2$ tests to investigate differences in proportions of losses or incomplete tracks between naive and experienced birds. For the latter, we ran one test per year to account for different GPS sampling regimes between 2019 and 2020.

## (g) Analyses of idiosyncratic route retention

To evaluate whether experienced birds remembered the specific routes they had developed in 2016, we measured to what extent homing paths resembled one another (i.e. 'path similarity'). Our main prediction was that if individuals remembered and re-used the route they had developed in 2016, their path in 2019–2020 would resemble more their own path(s) in 2016 (within-individual, between periods) than the path(s) of other individuals in 2019–2020 (between-individuals, within 2019–2020). To check that birds indeed had developed idiosyncratic routes in 2016, we checked that the path used in 2016 by an individual resembled more the other paths by the same individual in 2016 (within-individual, within-2016) than paths by different individuals in 2016 (between-individuals, within-2016). To test whether there was enough potential variation in homing routes in 2019–2020 when compared with 2016, we compared path-similarity 'between-individuals, within 2019–2020' with path-similarity 'between-individuals, within-2016'. Finally, to evaluate forgetting of the idiosyncratic route after several years, we compared path-similarity 'within-individual, between-periods' to path-similarity 'within-individual, within-2016'.

Similarity (or difference) between paths was determined as the average distance between paths (hereafter 'DBP'), calculated as the mean of the distances between each point on a focal path and the closest point on a reference path. The larger the DBP, the less similar are two paths. For these DBP calculations, we further down-sampled tracks to 1 point (location) every 10 s to save computing time. We then analysed route retention by individuals by means of Gaussian mixed models of log10-transformed DBP (a logarithmic scale was used for normality as DBP spanned several orders of magnitude). We retained only data from experienced individuals, and we further discarded incomplete flight paths from DBP analyses, which otherwise acted as uninformative outliers even on a logarithmic scale. To avoid artificially inflating the number of between-path comparisons because of the high number of path records per individual in 2016, we only used the last three paths recorded in 2016 for each individual (by which time homing routes are most likely to have stabilized [20,31,37]).

A first model included a four-level categorical variable ('within-individual, between-periods'; 'between-individuals, within 2019–20'; 'within-individual, within 2016'; and 'between-individual, within 2016'). This approach compared multiple paths by the same individual to multiple paths by the same or by other individuals. To control for this non-independence, we included as crossed-structured random effects the identity of the focal bird, the identity of the 'other' bird (sometimes equal to the focal identity for within-individual measures), as well as an identifier of the pair of birds involved (since multiple pairs of

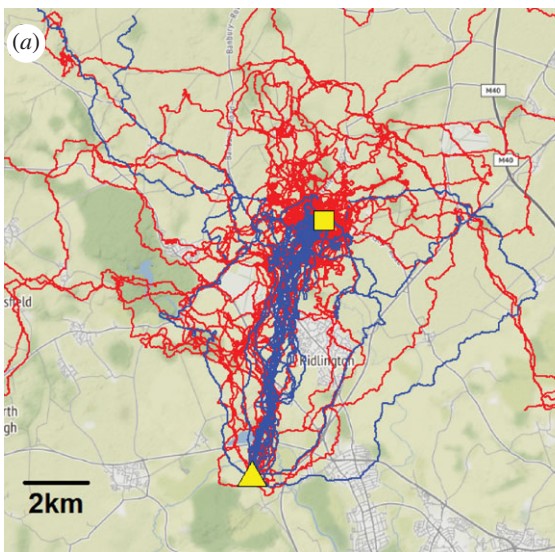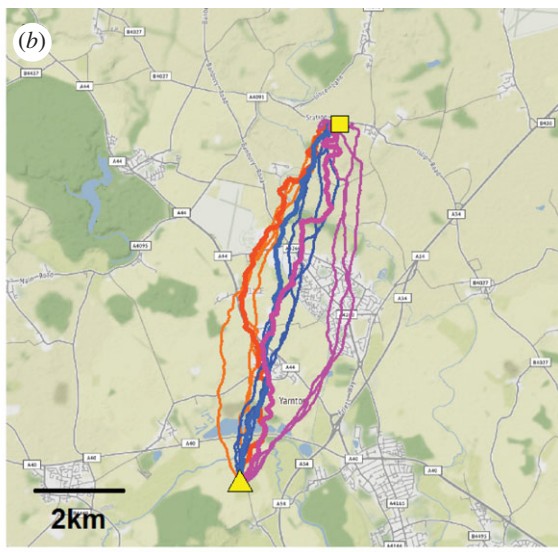

**Figure 1.** GPS records of pigeons' homing paths from the Greenhill Farm release site (yellow square) to their home loft at John Krebs Field Station (yellow triangle). (*a*) Comparisons of paths followed in 2019–2020 by naive pigeons (red tracks) and experienced pigeons (blue). Note some tracks extend beyond the map. (*b*) Example homing routes from three individuals (one colour per individual,) for which the last three paths in 2016 (thin lines) are compared to the path flown in 2019–2020 (thick lines). These three individuals were semi-randomly chosen on the basis of having dissimilar routes in 2016 (i.e. one westward, one eastward and one intermediate). Note that the spatial scales differ slightly between (*a*) and (*b*). © OpenStreetMap contributors. (Online version in colour.)

tracks were compared for each pair of birds). We used and report *post hoc* Tukey tests for pairwise comparisons between all four levels of the explanatory variable.

To analyse factors affecting the observed level of route retention by individuals, we ran a second mixed model investigating variations in 'within-individual, between-period' DBP only. We included two explanatory variables in this model, and bird identity as a random effect. First, to assess whether the social context during leaning affected route retention, we included a three-level categorical factor (individual, cultural or collective learning). Second, to account for potential inter-individual variations in the amount of forgetting since 2016, we included as a covariate an index of their homing efficiency (straightness, as described in the previous section) in their 2019–2020 release. We used *post hoc* Tukey tests for pairwise comparisons between learning treatments; for the effects of forgetting, we conservatively assumed the degrees of freedom to be the number of individuals ($n = 28$) rather than the number of DBP measures.

Finally, we investigated the persistence after several years of 'traditional signatures' in the homing routes used by different individuals belonging to the same cultural chain replicate (arising from cultural transmission of routes along the chain [37]). We ran a mixed-effect Gaussian linear model taking as a response variable the 'between-individual, within 2019–2020' DBP of only the pigeons from the cultural learning treatment. We included a two-level explanatory variable (individuals from the 'same' or from 'different' cultural transmission chain replicates) and the identity of the focal bird and the identity of the other bird as random effects. Again we made a conservative assumption for the degrees of freedom.

## (h) Software and packages
All analyses were carried out in the R environment v. 3.6. We used the package emmeans to run *post hoc* Tukey-adjusted pairwise comparisons. Mixed-models were built with package lme4. Maps were drawn with the OpenStreetMap package,

using open data made available under a CC-BY-SA 2.0 licence by the OpenStreetMap Foundation (OSMF).

## 3. Results

### (a) Data recovered, losses and incomplete tracks
From Greenhill Farm (figure 1*a*), we obtained and analysed homing paths for 33 birds in 2019 (seven experienced birds, and 26 naive birds) and 28 birds in 2020 (19 experienced birds and nine naive). We could compare these paths to tracks recorded at Greenhill Farm from the same individuals in 2016 (examples shown in figure 1*b*; detailed individual maps in electronic supplementary material, figure S3). From the control release site (Horspath), we obtained solo tracks from 25 birds in 2020 (18 birds with experience at Greenhill Farm and seven naive birds; electronic supplementary material, figure S1).

Not all pigeons succeeded in returning home; consistent with reliance on long-term memory, experienced birds were more likely to return from Greenhill Farm than naive birds (table 1: 31 out of 32 versus 40 out of 58: $\chi^2_1 = 8.042$, $p < 0.01$). At the control site (Horspath), only one naive bird never returned, the difference in proportion being non-significant (table 1; $\chi^2_1 = 0.189$, $p = 0.66$). Among the 61 birds that successfully returned from Greenhill Farm alone, 20 had incomplete tracks; this occurred even in 2020 when we modified the GPS setting to enable longer recording. Again consistent with a reliance on long-term memory, incomplete tracks from Greenhill Farm occurred mainly with naive birds, but the difference in proportion was not significant (complete tracks in exp. versus naive in 2019: 5 out of 7 versus 11 out of 26, $\chi^2_1 = 0.888$, $p = 0.35$; in 2020: 18 out of 19 versus 7 out of 9, $\chi^2_1 = 0.491$, $p = 0.48$). At Horspath, only experienced birds returned with incomplete tracks, but the

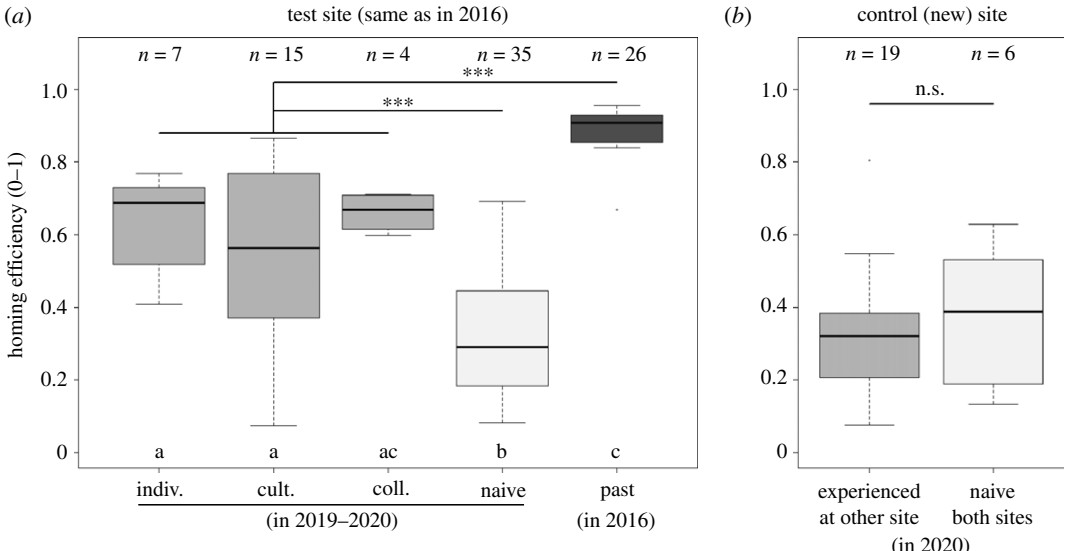

**Figure 2.** Homing efficiency measured as path straightness from (*a*) Greenhill Farm and (*b*) Horspath release sites. In (*a*), flights from experienced birds in 2019–2020 are broken down into distinct experimental treatments: 'indiv.': individual learning, 'cult.': cultural learning, and 'coll.': collective learning. Comparisons of efficiency between naive birds (light grey), experienced birds in 2019–2020 (all learning treatments pooled, medium grey) and experienced birds in 2016 (dark grey) are indicated (***: $p < 0.001$; n.s.: $p > 0.05$). Boxes sharing a common subscript letter were found not to differ statistically through *post hoc* Tukey tests in a model with a five-level categorical factor (significance at $p < 0.05$).

difference in proportion was non-significant (in 2020: 21 out of 23 versus 10 out of 10, $\chi_1^2 = 0.028$, $p = 0.87$).

## (b) Homing efficiency at Greenhill Farm (test release site)

We first compared homing efficiency (i.e. path straightness) from Greenhill Farm in 2019–2020 between naive and experienced pigeons (two-level factor) in a linear model also accounting for the year of release (two-level factor: 2019 or 2020). Experienced pigeons outperformed naive birds (figure 2*a*, $t_{58} = -4.529$, $p < 0.001$) with no difference detected between 2019 and 2020 ($t_{58} = -0.257$, $p = 0.80$). A second model evaluating forgetting relative to efficiency in 2016, run on experienced pigeons only (linear model with a two-level factor variable: 2019–2020 or 2016), revealed that the homing efficiency of experienced pigeons was lower in 2019–2020 than it was in the last flight of these individuals in 2016 (figure 2*a*, $t_{50} = -7.738$, $p < 0.001$).

To investigate the effect of the social context during learning on homing efficiency at Greenhill Farm 3 to 4 years later (figure 2*a*), we ran a model of straightness including a five-level categorical variable (naive, individual learning, cultural learning, collective learning, last flight in 2016), complemented by *post hoc* pairwise Tukey tests. Experienced pigeons from all learning treatments outperformed naive birds (naive versus indiv.: z = 4.586, $p < 0.001$; versus cult.: z = 4.365, $p < 0.001$; versus coll.: z = 4.042, $p < 0.001$). Pigeons from the individual and cultural learning treatments had lower efficiency than that observed in the last flight in 2016 (2016 versus indiv.: z = −3.862, $p = 0.001$; versus cult.: z = −6.774, $p < 0.001$); the difference was only marginally significant for the collective learning treatment (2016 versus coll.: z = −2.626, $p = 0.07$). There were no differences detected in homing efficiency in 2019–2020 between the different learning treatments (indiv. versus cult.: z = 1.206, $p = 0.75$; indiv. versus coll.: z = −0.374, $p = 0.99$; cult. versus coll.: z = −1.397, $p = 0.63$).

The original analysis of the 2016 dataset [37] reported that pigeons from the cultural learning treatment developed increasingly more efficient routes as the generations progressed (positive linear regression between efficiency and generation number). There was no such effect detected in the flights recorded in 2019–2020 from Greenhill Farm ($t_{18} = -1.448$, $p = 0.17$; electronic supplementary material, figure S2), but neither could we detect this effect in the subsample of 2016 data used for the present study ($t_{18} = -0.901$, $p = 0.38$).

## (c) Homing efficiency at Horspath (control release site)

At the control release site (Horspath), there was no difference in homing efficiency between birds with or without experience of the Greenhill Farm release site in 2016 (figure 2*b*, linear model with two-level categorical variable: $t_{23} = -0.045$, $p = 0.97$). This held true when testing each of the three learning context treatments separately against naive birds' efficiency (model with a four-level factor, with naive efficiency as baseline: individual learning: $t_{23} = -0.603$, $p = 0.55$; cultural learning: $t_{23} = 0.084$, $p = 0.94$; collective learning: $t_{23} = -0.072$, $p = 0.51$).

## (d) Retention of homing routes from Greenhill Farm

In 2016, individuals developed their own idiosyncratic routes, their last path resembling more the other paths flown by the same individual in 2016 than the paths of other individuals in 2016 (z = −8.567, $p < 0.001$; figure 2*a*). We found only a weak signal that individuals kept using their own idiosyncratic routes in 2019–2020 (figure 3*a*; electronic supplementary material, figure S3). Indeed, DBP was significantly lower within-individuals, between-periods (i.e. between 2016 and 2019–2020) than between-individuals, within 2019–2020 (z = 2.789, $p = 0.03$), but the magnitude of the difference was low (figure 3*a*). Furthermore, DBP within-individuals, between-periods was similar to DBP observed between-individuals, within 2016 (z = 1.351, $p = 0.53$), and much larger

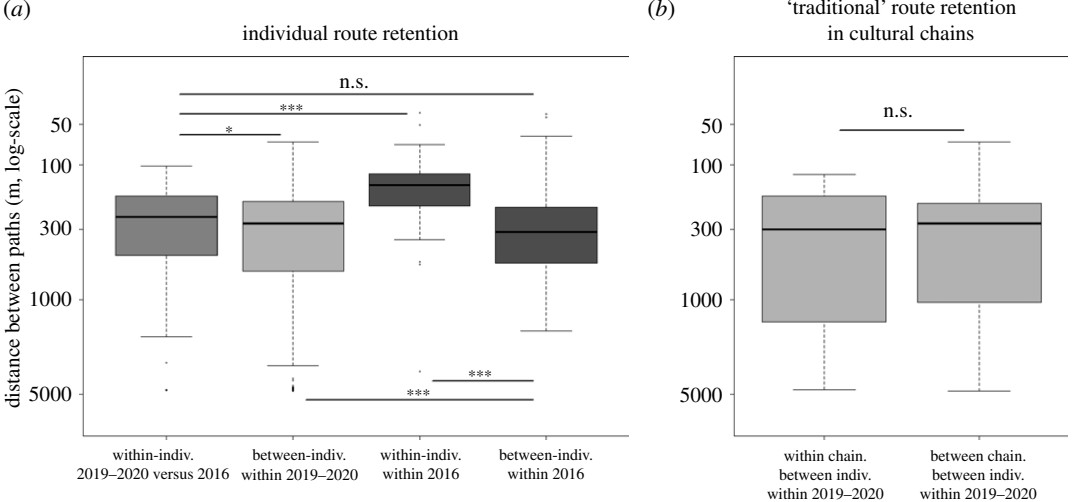

**Figure 3.** Route retention by an individual/chain evaluated from higher path similarity within-individual/chain than between-individual/chain. (*a*) Similarity in homing paths between or within individuals, between or within periods (intermediate grey between 2019–2020 and 2016; lighter grey within 2019–2020; darker grey within 2016; measured as distance between paths, or DBP (note 0 is at the top of the *y*-axis to have higher path similarity on top, scale is logarithmic). (*b*) DBP measured between individuals within 2019–2020, for pigeons of the cultural learning treatment, contrasted between individuals that belonged to the same cultural chain replicate (within chain) or to different chains (between chains).

than DBP observed within-individuals, within 2016 (figure 3*a*; $z = 8.307$, $p < 0.001$), indicative of forgetting. There was also significantly more between-individual path variation within 2019–2020 than within 2016 ($z = -3.902$, $p < 0.001$), although again the magnitude of the difference was low (figure 3*a*).

The social context during learning had no discernible effects on the level of within-individual path similarity between 2019–2020 and 2016 (indiv. versus cult.: $z = -0.798$, $p = 0.70$; indiv. versus coll.: $z = 0.285$, $p = 0.96$; cult. versus coll.: $z = 1.010$, $p = 0.57$). Within-individual path similarity between 2019–2020 and 2016 was higher in pigeons that showed higher homing efficiency in 2019–2020 ($t_{24} = -2.813$, $p < 0.01$; electronic supplementary material, figure S5). According to our mixed model, the predicted levels of within-individual, between-period DBP for pigeons with the highest homing efficiencies (straightness greater than 0.8, i.e. similar efficiency as the last flight in 2016, figure 2*a*) reached the average DBP levels observed within-individuals, within-2016, although with higher observed spread around the mean (electronic supplementary material, figure S5).

We found no signal for the persistence of 'traditional' route patterns in pigeons from the cultural learning treatment (figure 3*b*). In 2019–2020, between-individual path similarity was not significantly different when the two individuals had been part of different cultural transmission chain replicates versus when they had been part of the same cultural transmission chain replicate ($t_{18-305} = -0.341$, $p > 0.50$).

## 4. Discussion

Our results show that after 3–4 years without reinforcement, pigeons with extensive experience of a release site homed more efficiently than naive birds, only when released from this site but not when released from another site that all pigeons were naive to. This suggests that experienced pigeons could rely on memory acquired several years previously to efficiently solve a spatial task spanning *ca* 10 km over a natural landscape. However, at the scale of the

population, we found no clear evidence that experienced pigeons achieved this by flying along the same individual routes they had each established several years before.

The mechanisms underlying pigeon homing over familiar terrains have been extensively investigated, and reviewed in detail in [20]. Briefly, previous experiments, carried out on the scale of hours to weeks with regard to memory retention, suggest that each individual remembers and uses specific landmarks along its path to guide its journey home [20], presumably encoded in the hippocampal formation [21]. In particular, when pigeons are flying over familiar areas, they faithfully recapitulate idiosyncratic routes they had previously developed, even when they are released slightly 'off-course' [39], or when their solar compass is artificially shifted to deflect them by 90° from the correct orientation [40]. Here our results add novel insights on the role and scope of long-term memory in route navigation. While on average experienced birds clearly outperformed naive birds when homing from a known release site, on average they were also less faithful to their own routes than they had been several years before. Some individuals did show some striking similarity with their routes from several years before (electronic supplementary material, figures S3 and S5; e.g. birds S87 or A61); but others, even among those that homed efficiently along a relatively straight path, used a markedly different route than they had done previously (electronic supplementary material, figures S3 and S5; e.g. birds A53 or S13). In an important prior experiment, pigeons with lesioned hippocampal formations could still learn to home efficiently, but with lower path similarity than control birds [21]. This exemplifies the different mechanisms pigeons can rely on to navigate and to learn to home from a specific location (see [20,21] and references therein). It is thus unclear in our study whether experienced pigeons remembered some but not all specific landmarks along the way home (i.e. relied at least in part on pilotage), or simply the release site and a compass direction to home (i.e. used map-and-compass navigation), or even a combination of both, potentially variable across individuals. Their apparent partial forgetting could

also be linked to changes in the landscape itself. Still, pigeons experienced at one site did not outperform naive birds at another release site unfamiliar to all subjects, suggesting they could at least recognize and remember a specific release site, years after their last experience with it.

The social context during the initial learning did not appear to be a major source of variation in very long-term memory retention. Our sample size in each learning condition was too low to detect subtle differences in memory or forgetting between learning treatments; however, experienced pigeons from all learning treatments clearly outperformed naive pigeons. It seems therefore that memory retention on a very long-term time-scale is highly robust to social interactions occurring during learning in these pigeons. A further possibility that we cannot test here is that the social context during learning and the number of learning trials experienced by individuals had mutually compensating effects, as pigeons from the cultural learning treatment experienced fewer learning flights. We consider this unlikely, especially since we did not detect strong differences in homing behaviour between generation-1 and generation 2–4 birds that differed in social context during learning but not in total number of flights (electronic supplementary material, figure S2). As a mechanistic hypothesis, the lateralization of the pigeon brain may allow them to at least partially compartmentalize neural pathways involved in social interactions from those involved in navigation [41].

In our 2019–2020 data, we could no longer detect a 'traditional signature' in the homing routes of individuals that had been part of the same cultural transmission chain, as was observed in 2016 [37]. Neither did we detect the cumulative aspect of culture observed in 2016 [37]—that each new generation within a cultural transmission chain eventually outperformed the previous generation in homing efficiency—3 to 4 years later. Overall, individuals seemed no more likely to remember later over earlier paths (nor the opposite; electronic supplementary material, figure S4), even though in the cultural learning treatment these later paths with a tutee were on average closer to the straight-line home [37]. Again, our low sample size may have concealed such subtle quantitative effects. Alternatively, this may be owing to the partial forgetting of routes that we observed. These results suggest that the 'ratchet' of cumulative culture may not always be prevented from slipping backwards [42]—not only owing, as previously argued, to insufficient social transmission opportunities when population sizes/densities fall below a threshold [43], but also as a result of memory loss at the individual level if the task goes unreinforced for an extended period.

In summary, our results provide supporting evidence for memory of a spatial task persisting over several years,

irrespective of the social context during learning, even though individuals seemed not to remember all the landmarks they had previously recognized and used along the way. Such data are rare so it may be useful to reflect on how generalizable these results are. Domestic pigeons have long been recognized in captivity for their long-term memory capacities that can span years in picture discrimination [3,44] and aversive stimulus recognition and generalization [45,46]. Homing pigeon breeds have also been artificially selected for thousands of years for improved homing capacities [47], and the pigeons used in this experiment are from a population that has been under recent high selective pressure for homing success [20], although not directly on their very long-term memory capacities. In our population, pigeons are used to home both alone or in ever-changing groups, as this is part of their standard training procedure; this may have favoured strategies relying on individual learning robust to changing social contexts. All these factors may have contributed to the results we observed here, and limit their extrapolation. Yet our results do suggest that non-human animals can indeed remember, over multiple years, how to solve spatial tasks over real landscapes that they first experienced as adults. It lends further support but more nuanced cognitive mechanistic insights to previous anecdotal claims of very long-term spatial memories in wild animals and how such capacities may help them cope with extreme environmental events [17]. We believe that the comparative study of very long-term spatial memories offers promising and now accessible research avenues integrating ecology, evolution and neuroscience. Such integration may become increasingly important for predicting the behaviour of long-lived species under rapid global changes.

**Ethics.** Approval for work with pigeons was granted by the local Ethics committee of the Department of Zoology of the University of Oxford.

**Data accessibility.** Data and codes are available from the Dryad Digital Repository: https://doi.org/10.5061/dryad.qrfj6q5f7 [48].

**Authors' contributions.** J.C.: conceptualization, data curation, formal analysis, investigation, methodology, resources, software, visualization, writing—original draft, writing—review and editing; T.S.: conceptualization, data curation, resources, software, validation, writing—review and editing; D.B.: conceptualization, funding acquisition, project administration, resources, validation, writing—review and editing. All authors gave final approval for publication and agreed to be held accountable for the work performed therein.

**Competing interests.** We declare we have no competing interests.

**Funding.** Financial support was provided by the Templeton World Charity Foundation's 'Diverse Intelligences' scheme (grant no. TWCF0316 to D.B.).

**Acknowledgements.** We thank Lucy Larkman, David Wilson, Phil Smith and all John Krebs Field Station technicians for pigeon husbandry and technical support. We also thank two anonymous reviewers for constructive feedback on our manuscript.

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
