## [Peer Review File · Proceedings of the Royal Society B: Biological Sciences]

Review History

RSPB-2021-0655.R0 (Original submission)

Review form: Reviewer 1

Recommendation

Major revision is needed (please make suggestions in comments)

Scientific importance: Is the manuscript an original and important contribution to its field?

Good

General interest: Is the paper of sufficient general interest?

Good

Quality of the paper: Is the overall quality of the paper suitable?

Good

Is the length of the paper justified?

Yes

Should the paper be seen by a specialist statistical reviewer?

No

Do you have any concerns about statistical analyses in this paper? If so, please specify them explicitly in your report.

No

It is a condition of publication that authors make their supporting data, code and materials available - either as supplementary material or hosted in an external repository. Please rate, if applicable, the supporting data on the following criteria.

Is it accessible?

Yes

Is it clear?

Yes

Is it adequate?

Yes

Do you have any ethical concerns with this paper?

No

Comments to the Author

The manuscript addressed the interesting question of whether familiar, landmark-based spatial memory in homing pigeons persists over several years without any reinforcement, and whether social or solo learning may affect memory duration. The data reported suggest that pigeons remember previously taken routes, but social learning does not affect memory duration. I recommend a major revision of the manuscript based on the following concerns:

1) Introduction and Discussion. I find it curious that no mention was made of GPS tracking studies that have examined the strategies and navigational mechanisms used by homing pigeons to home from familiar locations. Is it more likely that the pigeons remember more the release site features and the compass direction leading them home, or the chain of landmarks followed after the learning of an idiosyncratic route? The authors might also consider a possible analysis to investigate this question.

2) Materials and Methods. The explanation of the treatments is difficult to follow. Maybe a flowchart or a table would help the reader better understand the treatments.

3) I have a major concern about the sample size of the experienced birds among the three groups, as they were very unbalanced. Does it make sense to consider a group of 4 birds (collective)? In addition, 7 birds against 35 does not make for a convincing comparison. I think it would be better to compare experienced and naïve birds without considering social learning experience. As written, the sample sizes do not allow for robust results and conclusions.

4) Figure 2. "Boxes sharing a common subscript letter were found not to differ statistically..." Is there a better way to report the results of the statistical comparisons in the Figure? I find this way not easily understandable.

5) Discussion "Our results indicate that individual pigeons remembered idiosyncratic homing routes they had learnt up to 4 years previously, even in the absence of any opportunity to fly these routes in the intervening period. Two key pieces of evidence support this conclusion. First, experimental birds' homing efficiency in 2019-2020 was significantly higher than that of naïve controls only at the site they had been released from 3-4 years earlier, but not at a second site, unfamiliar to all subjects." I do not agree with this statement. Birds might display a higher efficiency without remembering their idiosyncratic routes. Pigeons may not necessarily remember anything of about the chain of landmarks they followed several years earlier, but rather, simply remembered the association "release site-compass direction". It may also happen that pigeons display a high fidelity to their previous route even if these are not the most efficient.

6) Line 412 "and other avian brains". Maybe "other avian brain regions"? And in this context, what are the other brain regions involved in landmark-based navigation beyond the hippocampal formation?

Recently, a role of the hippocampal formation was reported for the learning of route fidelity

(Gagliardo et al 2020 Biol Lett.). As the authors discuss the neural basis of landmark-based navigation, it would seem appropriate to cite this work.

Review form: Reviewer 2

Recommendation

Major revision is needed (please make suggestions in comments)

Scientific importance: Is the manuscript an original and important contribution to its field?

Excellent

General interest: Is the paper of sufficient general interest?

Good

Quality of the paper: Is the overall quality of the paper suitable?

Marginal

Is the length of the paper justified?

Yes

Should the paper be seen by a specialist statistical reviewer?

No

Do you have any concerns about statistical analyses in this paper? If so, please specify them explicitly in your report.

Yes

It is a condition of publication that authors make their supporting data, code and materials available - either as supplementary material or hosted in an external repository. Please rate, if applicable, the supporting data on the following criteria.

Is it accessible?

Yes

Is it clear?

No

Is it adequate?

Yes

Do you have any ethical concerns with this paper?

No

Comments to the Author

This paper present analyses of homing behaviour in pigeons to test their memory for routes to which birds in a test group had already been previously exposed 3-4 years previous in comparison to birds naive to the specific homing task. The authors show that pigeons that had previously had opportunities to learn the route had better route efficiencie than those naive to the route, and that this was specifically due to the previous expsoure to that task since the experienced birds were no better than naive birds when both groups were exposed to an entirely new homing task. There was no evidence that the context of the initial learning experience - asocial, with another naive partner, or with an experience partner - had any effect on the efficacy of this memory, but in all three groups there was evidence of degraded efficiency compared to

the routes used at the end of the first exposure, suggesting that the memory degrades over time.

I think these are very interesting results and as the authors highlight it is challenging and therefore rare to study long term memory effects in any kinds of naturalistic tasks or settings so they are pretty valuable. However I have reservations about publishing this as is. I have one major comment and one minor for the authors to reflect on, and a number of issues with the statistical treatment, at least one part of which I am convinced is not correct. I also provide some specific minor comments as a list below.

The first, major, comment is the use of the term 'routes'. It is clear from the tracks presented that individual birds don't follow set 'routes' in the sense we would understand them from e.g. following a satnav or a set of directions (turn left here, then right etc). The paper subsumes all route information into one metric, the 'efficiency', or, specifically, the difference between the ground track of the birds and the straight line distance between the release point the the objective. The concept of a 'route' is quite a discrete one, whereas what is actually measured is a continuous aspect of the animal's movement. Given that we really can't be sure of what cues the birds are using and what navigational decisions they are making in real time, nor, beyond some general weather conditions, what the birds were exposed to during their flights, to what extent can we refer to these tracks as 'routes'? It implies a level of discreteness which I think just isn't supported in the data. For example, if a bird deviates from the straight line path because they detect some food, or to avoid a predator, to what extent is this a navigational decision about a route as opposed to a response to the conditions of the specific instance of navigation? It seems to me one can't. Therefore I question the use of the term 'route' in this context - it implies a kind of memory that we just can't speak to with these data, either way. Therefore I would ask the authors to reflect whether it might be more defensible, indeed more accurate, to refer simply to movement efficiency so for example their title might be 'Years-long memory effects on homing efficiency' to avoid what is from my perspective a somewhat problematic eliding of the straightforward concept of movement efficiency with the more tenuous implications of talking about discrete 'routes'.

The second comment is to question how specific are these results to the fact that the study animals were homing pigeons - were they in fact? Is there any need to discuss any possible effects of selection here given the competitive history of the pastime?

With respect to the statistical treatment there needs to be significant changes to how this is presented because as written it is impossible to understand the structure of the models that were fitted, nor assess how appropriate the model fit is - all we have are prose descriptions that are too vague (nobody could reconstruct the models from what is written) and single statistical results without context as to what other parameters or coefficients in the model are saying. There should be either precise descriptions (e.g. 'Route efficiency was modelled as a function of first route task experience (factor with levels experienced/naive) and test route task (factor with levels Greenhill/Horspath) assuming normally distributed errors'), formal equations or R syntax model formulae, and then full model summaries should be presented, and appropriate diagnostics included in the ESM. I have some concerns that while route efficiency is bounded between 0 and 1, the models appear to be of the normal error family, which has no such constraints on the predictions so I would like to be reassured that the models were not predicting impossible values ('predict' function in R) greater than 1 and less than 0, which would call into question the appropriateness of the models and suggest a different form was needed (for example, beta distributions which are constrained in the interval 0-1).

Secondly under this topic, the second analysis of 'nearest neighbour' distances (see comments below for discussion of this term), I noted the extremely high degrees of freedom in the test results reported, far exceeding the reported sample sizes (e.g. L295 df=9998). The methods report that this analysis calculated the 'mean distances between pairs of homing routes' but this df value exceeds even the pairwise n you would get if you compared 57 tracks to each other so it is very unclear how this analysis has been conducted to arrive at such a high df. I suspect that it is

possible the analysis has been done on individual pairs of points? In either case, the suggestion is either that each track has been entered into the analysis multiple times, or multiple points from each track have been analysed - in either case, this clearly violates the assumption of data independence (using the same track multiple times in a pairwise test, or using multiple points from the same track, which are surely heavily autocorrelated) and suggest to me that this analysis is not valid. The authors need to look again at this, and either describe their analysis with sufficient clarity that a reader can understand how df values of almost 10000 were obtained without violating independence assumptions, or perhaps consult with statisticians about how to model their data such that dependencies are accounted for (e.g. Mantel tests for pairwise comparisons, hierarchical models or GEEs to account for autocorrelation/data dependencies). I suspect that inflated certainty due to lack of accounting for dependence in the data is the reason we see such overlaps in Fig 3 while still obtaining p-values < 0.001). I cannot recommend publication until this concern is satisfied since it is fundamental to the author's interpretations.

Specific comments:

L33 this study refers to decades, not just 'several years'

L36 the unity of idea principle suggests this paragraph should be split here when the discussion of memory starts

L45 specifically, which domains?

L47 I thought this section of justification was weak - on what basis would the authors a priori expect that there might be differences?

L50 this would be more convincing if there were some methodology presented for the survey? What search terms were used etc the ESM is not clear on this

L58 the authors could consider whether reference to result on humpback migration (humpbacks <https://royalsocietypublishing.org/doi/full/10.1098/rsbl.2011.0279> <https://www.frontiersin.org/articles/10.3389/fmars.2020.00414>) are consistent with these assertions

L61 the quotes around 'true navigation' suggest this terms needs some explanation...

L80 I found this an awkward sentence

L109 naive 'to'?

L111-112 as an introduction, this paragraph loses focus here and starts to sound like the methods.. better to end with a clear statement of objectives and predicted outcomes/intepretations...

L118 the level of precision in these positions suggests that the study can measure things to the nearest 10cm, more or less. This seems unlikely - perhaps 3dp would suffice (nearest 100 metre, since the subsequent distance is reported to that level?)

L153 the sample size for collective learning seems small enough to question the value of including it

L158 do they always fly together? Presumably if birds in these pairs don't then social learning can't happen? Was this assessed?

L165 wouldn't 'for the purposes of analysis' be more accurate instead of 'in the Results section' here?

L176 I am confused as to how solo released birds could end up pairing up? It's a bit harsh on the reader to force them to park this thought and wait for pairing up criteria in a later section.... can't this be dealt with all together?

L186 I don't quite get this section - could it be clarified and its rationale/significance explained?

L193 I need some more explanation here as to why this asumption of straight line return following GPS failure is conservative with respect to the analysese reported here?

L204 I think 'straight line' would be more cross-culturally accessible here

L209 the term 'nearest neighbour distance' is I think pretty suboptimal here - nearly every animal behaviour researchers, and I suspect a lot of others as well, will intuitively read this as relating to a literal nearest neighbour - this is what the term is used for in virtually every other context I have seen it in the literature, meaning the distance to another animal, when as I undertand it is nothing of the sort, but instead the distance between a given GPS fix in a focal track to the nearest point on a reference track. Please rethink this terminology

L211 I think we need more clarification of the selection criteria here

L220 should give model formulation - unclear here how models might be predicted - and identify key predictions/parameters. Also, response is in the range 0-1 - how often did gaussian models predict 'efficiencies' outside this range? I tried to look at the code but found it too poorly commented to properly assess this

L225 again model descriptions are not detailed enough

L228 I found this sentence super confusing, I could not understand it - please attempt to clarify

L230 'explored' is too vague here - did you identify the the most similar route? to what end?

L254 maybe it is the rendering by the journal but this figure appeared low quality and the text was only just big enough to read, probably would not survive reduction in formatting to column width

L254 Also I don't understand why Tukey pairwise tests were needed when a model which had 'Naive' as a baseline in a 'treatment' factor with levels Naive/Indiv/Cult/Coll/Final2016 would not suffice?

L263 it is too unclear specifically what is being tested here for interpretation to really be possible for a naive reader (like me)

L271 wasn't this already stated above?

L306 I struggled to understand this sentence please clarify

L317 this is where the eliding of efficiency and route memory really bites - define 'remembered' (seems binary but results suggest gradual decline?)/ What is likely to be the thing(s) that are remembered?

L327 could the authors provide any evidence of landmark use from the tracks in this study?

L328 Really dislike claims of primacy/superiority like this - 'most robust' is vague enough to bring this close to rhetorical anyway. Suggest removing.

L343 I think the focus on collective learning is excessive here given the sample size

L403 reference to work by Carroll and colleagues on whale populations struggling to recover migration routes might be pertinent here? <https://www.nature.com/articles/srep16182>

Decision letter (RSPB-2021-0655.R0)

20-Apr-2021

Dear Mr Collet:

I am writing to inform you that your manuscript RSPB-2021-0655 entitled "Years-long memory of individually, culturally and collectively learnt homing routes in pigeons" has, in its current form, been rejected for publication in Proceedings B.

This action has been taken on the advice of referees, who have recommended that substantial revisions are necessary. With this in mind we would be happy to consider a resubmission, provided the comments of the referees are fully addressed. However please note that this is not a provisional acceptance.

Sincerely,
 Dr Robert Barton
 mailto: proceedingsb@royalsociety.org

Associate Editor

Comments to Author:

This article tackles an interesting question but relies on a somewhat messy data set. While the use of historic and patchy data cannot be avoided, I do agree with the two reviewers that the methods could be revised to enhance the clarity of their presentation. As written, it is very difficult to keep track of all the different treatment groups and how and when the tests were administered. Additionally, while both reviewers note the value of this dataset, given the scarcity of such data, the reviewers also highlight concerns with how the data were treated and analyzed - both in terms of the clarity of analysis presentation and how the analysis was administered. Therefore, at this time, it is difficult to fully evaluate the merit of this study. Finally, reviewer 2 proposes new frameworks as to how the pigeons' behavior could be considered and described and I believe that these discussions are valuable.

Reviewer(s)' Comments to Author:

Referee: 1

Comments to the Author(s)

The manuscript addressed the interesting question of whether familiar, landmark-based spatial memory in homing pigeons persists over several years without any reinforcement, and whether social or solo learning may affect memory duration. The data reported suggest that pigeons remember previously taken routes, but social learning does not affect memory duration. I recommend a major revision of the manuscript based on the following concerns:

- 1) Introduction and Discussion. I find it curious that no mention was made of GPS tracking studies that have examined the strategies and navigational mechanisms used by homing pigeons to home from familiar locations. Is it more likely that the pigeons remember more the release site features and the compass direction leading them home, or the chain of landmarks followed after the learning of an idiosyncratic route? The authors might also consider a possible analysis to investigate this question.
- 2) Materials and Methods. The explanation of the treatments is difficult to follow. Maybe a flowchart or a table would help the reader better understand the treatments.
- 3) I have a major concern about the sample size of the experienced birds among the three groups, as they were very unbalanced. Does it make sense to consider a group of 4 birds (collective)? In addition, 7 birds against 35 does not make for a convincing comparison. I think it would be better to compare experienced and naïve birds without considering social learning experience. As written, the sample sizes do not allow for robust results and conclusions.
- 4) Figure 2. "Boxes sharing a common subscript letter were found not to differ statistically..." Is there a better way to report the results of the statistical comparisons in the Figure? I find this way not easily understandable.

5) Discussion “Our results indicate that individual pigeons remembered idiosyncratic homing routes they had learnt up to 4 years previously, even in the absence of any opportunity to fly these routes in the intervening period. Two key pieces of evidence support this conclusion. First, experimental birds’ homing efficiency in 2019-2020 was significantly higher than that of naïve controls only at the site they had been released from 3-4 years earlier, but not at a second site, unfamiliar to all subjects.” I do not agree with this statement. Birds might display a higher efficiency without remembering their idiosyncratic routes. Pigeons may not necessarily remember anything of about the chain of landmarks they followed several years earlier, but rather, simply remembered the association “release site-compass direction”. It may also happen that pigeons display a high fidelity to their previous route even if these are not the most efficient.

6) Line 412 “and other avian brains”. Maybe “other avian brain regions”? And in this context, what are the other brain regions involved in landmark-based navigation beyond the hippocampal formation?

Recently, a role of the hippocampal formation was reported for the learning of route fidelity (Gagliardo et al 2020 Biol Lett.). As the authors discuss the neural basis of landmark-based navigation, it would seem appropriate to cite this work.

Referee: 2

Comments to the Author(s)

This paper present analyses of homing behaviour in pigeons to test their memory for routes to which birds in a test group had already been previously exposed 3-4 years previous in comparison to birds naive to the specific homing task. The authors show that pigeons that had previously had opportunities to learn the route had better route efficiency than those naive to the route, and that this was specifically due to the previous exposure to that task since the experienced birds were no better than naive birds when both groups were exposed to an entirely new homing task. There was no evidence that the context of the initial learning experience - asocial, with another naive partner, or with an experience partner - had any effect on the efficacy of this memory, but in all three groups there was evidence of degraded efficiency compared to the routes used at the end of the first exposure, suggesting that the memory degrades over time.

I think these are very interesting results and as the authors highlight it is challenging and therefore rare to study long term memory effects in any kinds of naturalistic tasks or settings so they are pretty valuable. However I have reservations about publishing this as is. I have one major comment and one minor for the authors to reflect on, and a number of issues with the statistical treatment, at least one part of which I am convinced is not correct. I also provide some specific minor comments as a list below.

The first, major, comment is the use of the term 'routes'. It is clear from the tracks presented that individual birds don't follow set 'routes' in the sense we would understand them from e.g. following a satnav or a set of directions (turn left here, then right etc). The paper subsumes all route information into one metric, the 'efficiency', or, specifically, the difference between the ground track of the birds and the straight line distance between the release point the the objective. The concept of a 'route' is quite a discrete one, whereas what is actually measured is a continuous aspect of the animal's movement. Given that we really can't be sure of what cues the birds are using and what navigational decisions they are making in real time, nor, beyond some general weather conditions, what the birds were exposed to during their flights, to what extent can we refer to these tracks as 'routes'? It implies a level of discreteness which I think just isn't supported in the data. For example, if a bird deviates from the straight line path because they detect some food, or to avoid a predator, to what extent is this a navigational decision about a route as opposed to a response to the conditions of the specific instance of navigation? It seems to me one can't. Therefore I question the use of the term 'route' in this context - it implies a kind of memory that we just can't speak to with these data, either way. Therefore I would ask the authors to reflect whether it might be more defensible, indeed more accurate, to refer simply to movement efficiency so for example their title might be 'Years-long memory effects on homing efficiency' to avoid what is from my perspective a somewhat problematic eliding of the

straightforward concept of movement efficiency with the more tenuous implications of talking about discrete 'routes'.

The second comment is to question how specific are these results to the fact that the study animals were homing pigeons - were they in fact? Is there any need to discuss any possible effects of selection here given the competitive history of the pastime?

With respect to the statistical treatment there needs to be significant changes to how this is presented because as written it is impossible to understand the structure of the models that were fitted, nor assess how appropriate the model fit is - all we have are prose descriptions that are too vague (nobody could reconstruct the models from what is written) and single statistical results without context as to what other parameters or coefficients in the model are saying. There should be either precise descriptions (e.g. 'Route efficiency was modelled as a function of first route task experience (factor with levels experienced/naive) and test route task (factor with levels Greenhill/Horspath) assuming normally distributed errors'), formal equations or R syntax model formulae, and then full model summaries should be presented, and appropriate diagnostics included in the ESM. I have some concerns that while route efficiency is bounded between 0 and 1, the models appear to be of the normal error family, which has no such constraints on the predictions so I would like to be reassured that the models were not predicting impossible values ('predict' function in R) greater than 1 and less than 0, which would call into question the appropriateness of the models and suggest a different form was needed (for example, beta distributions which are constrained in the interval 0-1).

Secondly under this topic, the second analysis of 'nearest neighbour' distances (see comments below for discussion of this term), I noted the extremely high degrees of freedom in the test results reported, far exceeding the reported sample sizes (e.g. L295 df=9998). The methods report that this analysis calculated the 'mean distances between pairs of homing routes' but this df value exceeds even the pairwise n you would get if you compared 57 tracks to each other so it is very unclear how this analysis has been conducted to arrive at such a high df. I suspect that it is possible the analysis has been done on individual pairs of points? In either case, the suggestion is either that each track has been entered into the analysis multiple times, or multiple points from each track have been analysed - in either case, this clearly violates the assumption of data independence (using the same track multiple times in a pairwise test, or using multiple points from the same track, which are surely heavily autocorrelated) and suggest to me that this analysis is not valid. The authors need to look again at this, and either describe their analysis with sufficient clarity that a reader can understand how df values of almost 10000 were obtained without violating independence assumptions, or perhaps consult with statisticians about how to model their data such that dependencies are accounted for (e.g. Mantel tests for pairwise comparisons, hierarchical models or GEEs to account for autocorrelation/data dependencies). I suspect that inflated certainty due to lack of accounting for dependence in the data is the reason we see such overlaps in Fig 3 while still obtaining p-values < 0.001). I cannot recommend publication until this concern is satisfied since it is fundamental to the author's interpretations.

Specific comments:

L33 this study refers to decades, not just 'several years'

L36 the unity of idea principle suggests this paragraph should be split here when the discussion of memory starts

L45 specifically, which domains?

L47 I thought this section of justification was weak - on what basis would the authors a priori expect that there might be differences?

L50 this would be more convincing if there were some methodology presented for the survey?

What search terms were used etc the ESM is not clear on this

L58 the authors could consider whether reference to result on humpback migration (humpbacks <https://royalsocietypublishing.org/doi/full/10.1098/rsbl.2011.0279>)

<https://www.frontiersin.org/articles/10.3389/fmars.2020.00414>) are consistent with these assertions

L61 the quotes around 'true navigation' suggest this terms needs some explanation...

L80 I found this an awkward sentence

L109 naive 'to'?

L111-112 as an introduction, this paragraph loses focus here and starts to sound like the methods.. better to end with a clear statement of objectives and predicted outcomes/intepretations...

L118 the level of precision in these positions suggests that the study can measure things to the nearest 10cm, more or less. This seems unlikely - perhaps 3dp would suffice (nearest 100 metre, since the subsequent distance is reported to that level?)

L153 the sample size for collective learning seems small enough to question the value of including it

L158 do they always fly together? Presumably if birds in these pairs don't then social learning can't happen? Was this assessed?

L165 wouldn't 'for the purposes of analysis' be more accurate instead of 'in the Results section' here?

L176 I am confused as to how solo released birds could end up pairing up? It's a bit harsh on the reader to force them to park this thought and wait for pairing up criteria in a later section.... can't this be dealt with all together?

L186 I don't quite get this section - could it be clarified and its rationale/significance explained?

L193 I need some more explanation here as to why this asumption of straight line return following GPS failure is conservative with respect to the analysese reported here?

L204 I think 'straight line' would be more cross-culturally accessible here

L209 the term 'nearest neighbour distance' is I think pretty suboptimal here - nearly every animal behaviour researchers, and I suspect a lot of others as well, will intuitively read this as relating to a literal nearest neighbour - this is what the term is used for in virtually every other context I have seen it in the literature, meaning the distance to another animal, when as I undertand it is nothing of the sort, but instead the distance between a given GPS fix in a focal track to the nearest point on a reference track. Please rethink this terminology

L211 I think we need more clarification of the selection criteria here

L220 shoud give model formulation - unclear here how models might be predicted - and identify key predictions/parameters. Also, response is in the range 0-1 - how often did guassian models predict 'efficiencies' outside this range? I tried to look at the code but found it too poorly commented to properly assess this

L225 again model descriptions are not detailed enough

L228 I found this sentence super confusing, I could not understand it - please attempt to clarify

L230 'explored' is too vague here - did you identify the the most similar route? to what end?

L254 maybe it is the rendering by the journal but this figure appeared low quality and the text was only just big enough to read, probably would not survive reduction in formatting to column width

L254 Also I don't understand why Tueky pariwise tests were needed when a model which had 'Naive' as a baseline in a 'treatment' factor with levels Naive/Indiv/Cult/Coll/Final2016 would not suffice?

L263 it is too unclear specifically what is being tested here for intepretation to really be possible for a naive reader (like me)

L271 wasn't this already stated above?

L306 I struggled to understand this sentence please clarify

L317 this is where the eliding of efficiency and route memory really bites - define 'remembered' (seems binary but results suggest gradual decline?)/ What is likely to be the thing(s) that are remembered?

L327 could the authors provide any evidence of landmark use from the tracks in this study?

L328 Really dislike claims of primacy/superiority like this - 'most robust' is vague enough to bring this close to rhetorical anyway. Suggest removing.

L343 I think the focus on collective learning is excessive here given the sample size

L403 reference to work by Carroll and colleagues on whale populations struggling to recover migration routes might be pertinent here? <https://www.nature.com/articles/srep16182>

Author's Response to Decision Letter for (RSPB-2021-0655.R0)

See Appendix A.

RSPB-2021-2110.R0 (Revision)

Review form: Reviewer 2

Recommendation

Major revision is needed (please make suggestions in comments)

Scientific importance: Is the manuscript an original and important contribution to its field?

Excellent

General interest: Is the paper of sufficient general interest?

Good

Quality of the paper: Is the overall quality of the paper suitable?

Excellent

Is the length of the paper justified?

Yes

Should the paper be seen by a specialist statistical reviewer?

No

Do you have any concerns about statistical analyses in this paper? If so, please specify them explicitly in your report.

No

It is a condition of publication that authors make their supporting data, code and materials available - either as supplementary material or hosted in an external repository. Please rate, if applicable, the supporting data on the following criteria.

Is it accessible?

Yes

Is it clear?

Yes

Is it adequate?

Yes

Do you have any ethical concerns with this paper?

No

Comments to the Author

I am content that the authors have made excellent efforts to address the comments I made on the original draft, and I think the manuscript is markedly improved by those efforts. In my view it is now ready to be published and should make a very interesting addition to the literature. I have no further comments or concerns.

Decision letter (RSPB-2021-2110.R0)

11-Oct-2021

Dear Mr Collet:

Your manuscript has now been peer reviewed and the reviews have been assessed by an Associate Editor. The reviewers' comments (not including confidential comments to the Editor) and the comments from the Associate Editor are included at the end of this email for your reference. As you will see, the reviewers and the Editors have raised some concerns with your manuscript and we would like to invite you to revise your manuscript to address them.

Research ethics:

Use of animals and field studies:

It is a condition of publication that you make available the data and research materials supporting the results in the article (<https://royalsociety.org/journals/authors/author-guidelines/#data>). Datasets should be deposited in an appropriate publicly available repository and details of the associated accession number, link or DOI to the datasets must be included in the Data Accessibility section of the article (<https://royalsociety.org/journals/ethics-policies/data-sharing-mining/>). Reference(s) to datasets should also be included in the reference list of the article with DOIs (where available).

Please submit a copy of your revised paper within three weeks. If we do not hear from you within this time your manuscript will be rejected. If you are unable to meet this deadline please let us know as soon as possible, as we may be able to grant a short extension.

Best wishes,

Dr Robert Barton

Associate Editor

Comments to Author:

One of the reviewers who reviewed your original submission has reviewed this revision, as have I. I want to thank you for the careful revisions that you have made to your reporting. It is now much more clear - especially the methods and results. I believe that this study will be of great value to the field.

However, as you worked with captive animals, I request that you provide a more detailed description on the care and housing protocols for these birds outside of test times e.g., please describe the aviary, the food and water schedule, and other pertinent details about their husbandry. See the ARRIVE 2.0 guidelines for a template. This information can be provided in your supplementary materials, but is important to report to ensure rigor and reproducibility. Especially important for a study such as this would be to know the diet of the birds, and if there were any weight or body condition requirements for inclusion in the study.

Percie du Sert N, Hurst V, Ahluwalia A, Alam S, Avey MT, Baker M, et al. (2020) The ARRIVE guidelines 2.0: Updated guidelines for reporting animal research. *PLoS Biol* 18(7): e3000410. <https://doi.org/10.1371/journal.pbio.3000410>

Reviewer(s)' Comments to Author:

Referee: 2

Comments to the Author(s).

I am content that the authors have made excellent efforts to address the comments I made on the original draft, and I think the manuscript is markedly improved by those efforts. In my view it is now ready to be published and should make a very interesting addition to the literature. I have no further comments or concerns.

Author's Response to Decision Letter for (RSPB-2021-2110.R0)

See Appendix B.

Decision letter (RSPB-2021-2110.R1)

22-Oct-2021

Dear Mr Collet

I am pleased to inform you that your manuscript entitled "Pigeons retain partial memories of homing paths years after learning them individually, collectively or culturally" has been accepted for publication in *Proceedings B*.

Data Accessibility section

Open Access

Paper charges

Sincerely,

Dr Robert Barton

Associate Editor:

Comments to Author:

Thank you for your quick and thorough response to my request. The in-text and supplemental materials fully provide all the information I requested.

Appendix A

RSPB-2021-0655

Response to reviewers' comments

We thank the editors and reviewers for carefully considering our manuscript and for their constructive remarks, copy-pasted below. We followed virtually all suggestions, leading to substantial text edits in all parts of the manuscript. Our point-by-point answers to comments below appear in bold. Line numbers in referees' comments refer to the original submission, line numbers in our answers refer to the clean revised version.

Associate Editor

Comments to Author:

This article tackles an interesting question but relies on a somewhat messy data set. While the use of historic and patchy data cannot be avoided, I do agree with the two reviewers that the methods could be revised to enhance the clarity of their presentation. As written, it is very difficult to keep track of all the different treatment groups and how and when the tests were administered.

We have entirely re-written and re-organized the M&M section, and we now provide a table summarizing the different treatments and corresponding sample sizes. Statistical models are now described in much more detail (l.208-215, l.245-270, l.312-336). To compensate for the increase in the length of the manuscript, we simplified some parts of the discussion and reduced the number of references.

Additionally, while both reviewers note the value of this dataset, given the scarcity of such data, the reviewers also highlight concerns with how the data were treated and analyzed - both in terms of the clarity of analysis presentation and how the analysis was administered. Therefore, at this time, it is difficult to fully evaluate the merit of this study.

We have clarified the methods and models used for analyses (see answer above), we adopted a more sophisticated and more robust statistical treatment for route retention analyses (mixed-effect model) as suggested by reviewer 2 (l.248-251), and added new information in the ESM (detailed individual maps, correlation between homing efficiency and level of route retention). This did not fundamentally change our overall conclusions (partial forgetting of routes by individuals), but we reformulated our interpretation (and title) to more accurately reflect the uncertainty about whether pigeons remember part of the specific routes or something else (l.379-409).

Finally, reviewer 2 proposes new frameworks as to how the pigeons' behavior could be considered and described and I believe that these discussions are valuable.

Following the reviewers' suggestions, we have considerably developed and clarified the concept of routes and how pigeons are known to navigate over familiar terrains, in the introduction (l.60, l.83-87) but also and mainly in the discussion (l. 386-409) to provide a more tractable and more accurate interpretation of our results accessible to any reader. We believe this considerably improved the manuscript indeed, thank you.

Reviewer(s)' Comments to Author:

Referee: 1

Comments to the Author(s)

The manuscript addressed the interesting question of whether familiar, landmark-based spatial memory in homing pigeons persists over several years without any reinforcement, and whether social or solo learning may affect memory duration. The data reported suggest that pigeons remember previously taken routes, but social learning does not affect memory duration. I recommend a major revision of the manuscript based on the following concerns:

1) Introduction and Discussion. I find it curious that no mention was made of GPS tracking studies that have examined the strategies and navigational mechanisms used by homing pigeons to home from familiar locations. Is it more likely that the pigeons remember more the release site features and the compass direction leading them home, or the chain of landmarks followed after the learning of an idiosyncratic route? The authors might also consider a possible analysis to investigate this question.

We now mention in the introduction this previous literature on pigeons and the conclusions as to pigeons' use of landmarks and routes (l.60, l.83-87). In the discussion we come back to the main arguments that led previous studies to conclude reliance on landmarks and routes, to better reflect on our own results and their interpretation (l.386-409). These key arguments were either higher within-individual than between individual route resemblance, as reported here, or coming from experiments that could not be replicated for the present study (e.g. sun-compass manipulation, hippocampal ablations, etc.). However we did include an additional analysis (correlation between observed route retention and observed homing efficiency, l. 257-259, l. 366-372, new supplementary figure S5) as well as maps of paths for each individual (ESM Fig. S3) to better emphasize that it is difficult to conclude firmly in our study whether pigeons have completely forgotten the routes or only partially, and if this varies across individuals (see discussion l. 396-399).

2) Materials and Methods. The explanation of the treatments is difficult to follow. Maybe a flowchart or a table would help the reader better understand the treatments.

We reorganized our M&M section to better explain the treatments and the share of arbitrary choices it implied for our analyses, and we included a table (Table 1) to help follow the complex sample size and data structure.

3) I have a major concern about the sample size of the experienced birds among the three groups, as they were very unbalanced. Does it make sense to consider a group of 4 birds (collective)? In addition, 7 birds against 35 does not make for a convincing comparison. I think it would be better to compare experienced and naïve birds without considering social learning experience. As written, the sample sizes do not allow for robust results and conclusions.

We hope to have clarified that we followed a two-step analysis (for both homing efficiency and path resemblance analyses): first compare experienced vs naïve as you propose; then break it down into different social context categories. While we fully agree that our sample sizes are low and involve some arbitrary categorization, we believe it would be a shame not to report such rare data; besides, our post-hoc tests show that the sample size is sufficient to contrast each social context treatment with naïve birds, which we aim as our main message for the social context questions (now hopefully better reflected in the discussion). We have increased the emphasis on our low sample size for social context questions in the discussion (l.410-413).

4) Figure 2. "Boxes sharing a common subscript letter were found not to differ statistically..." Is there a better way to report the results of the statistical comparisons in the Figure? I find this way not easily understandable.

We kept these subscript letters as we can't think of a better way to represent all post-hoc pairwise comparisons in a readable manner, but we added more classical comparative lines to reflect the first step of our analyses (experienced vs naïve, regardless of social context).

5) Discussion "Our results indicate that individual pigeons remembered idiosyncratic homing routes they had learnt up to 4 years previously, even in the absence of any opportunity to fly these routes in the intervening period. Two key pieces of evidence support this conclusion. First, experimental birds' homing efficiency in 2019-2020 was significantly higher than that of naïve controls only at the site they had been released from 3-4 years earlier, but not at a second site, unfamiliar to all subjects." I do not agree with this statement. Birds might display a higher efficiency without remembering their idiosyncratic routes. Pigeons may not necessarily remember anything of about the chain of landmarks they followed several years earlier, but rather, simply remembered the association "release site-compass direction". It may also happen that pigeons display a high fidelity to their previous route even if these are not the most efficient.

We agree with you and we have largely edited the discussion and abstract to better explain this, in light of previous results from pigeon homing experiments (l. 379-409). We also changed and simplified the title.

6) Line 412 "and other avian brains". Maybe "other avian brain regions"? And in this context, what are the other brain regions involved in landmark-based navigation beyond the hippocampal formation?

Recently, a role of the hippocampal formation was reported for the learning of route fidelity (Gagliardo et al 2020 Biol Lett.). As the authors discuss the neural basis of landmark-based navigation, it would seem appropriate to cite this work.

We have added this reference and a discussion of its significance for our results (l.390, l. 399-401), thank you. We think this has clarified the role of the hippocampal formation in landmark memory but not in other, poorly understood back-up mechanisms for pigeon navigation over familiar terrains (l.402-406).

Referee: 2

Comments to the Author(s)

This paper presents analyses of homing behaviour in pigeons to test their memory for routes to which birds in a test group had already been previously exposed 3-4 years previous in comparison to birds naive to the specific homing task. The authors show that pigeons that had previously had opportunities to learn the route had better route efficiency than those naive to the route, and that this was specifically due to the previous exposure to that task since the experienced birds were no better than naive birds when both groups were exposed to an entirely new homing task. There was no evidence that the context of the initial learning experience - asocial, with another naive partner, or with an experience partner - had any effect on the efficacy of this memory, but in all three groups there was evidence of degraded efficiency compared to the routes used at the end of the first exposure, suggesting that the memory degrades over time.

I think these are very interesting results and as the authors highlight it is challenging and therefore rare to study long term memory effects in any kinds of naturalistic tasks or settings so they are pretty valuable. However I have reservations about publishing this as is. I have one major comment and one minor for the authors to reflect on, and a number of issues with the statistical treatment, at least one part of which I am convinced is not correct. I also provide some specific minor comments as a list below.

The first, major, comment is the use of the term 'routes'. It is clear from the tracks presented that individual birds don't follow set 'routes' in the sense we would understand them from e.g. following a satnav or a set of directions (turn left here, then right etc).

We have clarified why we consider that pigeons usually rely on following "routes" over familiar terrains (i.e. remember chains of landmarks), based on extensive previous literature in homing pigeons (l.60, l.83-87, l.386-409). We have also clarified that our evidence is equivocal as to whether they still remember these routes, at least partially, 3 to 4 years later (l.401-406).

The paper subsumes all route information into one metric, the 'efficiency', or, specifically, the difference between the ground track of the birds and the straight line distance between the release point and the objective.

We agree that homing efficiency/straightness is not a measure of route use. We changed subsection headings in M&M and Results and we clarified throughout that our discussions on routes are mainly arising from the observation that paths flown by the same individual are more similar to each other than they are to paths flown by other individuals. This same observation has been used by many previous papers (from our lab as well as from others) to argue that pigeons perform route-based navigation (also reflected in various terms used in the same literature: route learning, route following, route fidelity, route loyalty).

The concept of a 'route' is quite a discrete one, whereas what is actually measured is a continuous aspect of the animal's movement.

We respectfully disagree with this: in short-time-scale experiments most if not all individuals show striking fidelity to a narrow corridor across successive releases (reviewed in our reference 20), although there is variance in the width of these corridors and the corridors of different individuals can partly overlap with each other.

Given that we really can't be sure of what cues the birds are using and what navigational decisions they are making in real time, nor, beyond some general weather conditions, what the birds were exposed to during their flights, to what extent can we refer to these tracks as 'routes'?

We now include a more detailed discussion of the previous literature on pigeon homing that is key for our interpretation of the tracks as 'routes' (l.386-409). Apologies that this was missing before, as it is indeed key to understanding our results and interpretations.

It implies a level of discreteness which I think just isn't supported in the data.

We partly disagree with this as we do find a significant signal, albeit weak and of low magnitude, for paths flown by individuals in 2019-20 resembling more their own path from 2016 than the paths flown by other individuals in 2019-20 (l.356-359, Fig.3A). We have tried to clarify this argumentation throughout the manuscript.

For example, if a bird deviates from the straight line path because they detect some food, or to avoid a predator, to what extent is this a navigational decision about a route as opposed to a response to the conditions of the specific instance of navigation? It seems to me one can't.

This possibility is not supported by the extensive previous literature on homing by pigeons once they establish local familiarity with the landscape, now better detailed in the manuscript (l.386-409), and also does not reflect our significant-but-weak signal in Figure 3 as stated in our immediately preceding answer.

Therefore I question the use of the term 'route' in this context - it implies a kind of memory that we just can't speak to with these data, either way.

We have tried to be more careful with our use of "route"; we acknowledge it previously conflated previous literature knowledge with our own observations. We now try to use "route" only when we can see more between-path resemblance within individuals than between individuals (implying that routes are not single tracks from one release experiment but properties observed from many tracks and consecutive releases), and to use "path" to refer to records of a single homing track from a single release. Note that this is consistent with terminology used, beyond pigeons, in a lot of avian movement literature, or in desert ant navigation for instance.

Therefore I would ask the authors to reflect whether it might be more defensible, indeed more accurate, to refer simply to movement efficiency so for example their title might be 'Years-long memory effects on homing efficiency' to avoid what is from my perspective a somewhat problematic eliding of the straightforward concept of movement efficiency with the more tenuous implications of talking about discrete 'routes'.

Given the complexity and nuances behind the word "route" we agree and we changed the title to avoid including "route" and better reflect nuances in our conclusions.

The second comment is to question how specific are these results to the fact that the study animals were homing pigeons - were they in fact? Is there any need to discuss any possible effects of selection here given the competitive history of the pastime?

It is difficult to find a clear definition of “homing pigeons” and/or to ascertain that our population would come from a pure lineage of such a clade if it exists. However, we do include in our discussion a reflection on the fact that this population has a long history (~30 years) of experiments and therefore selection for homing capacities (l.443-446). We stress however that it is unlikely that any homing pigeon population has ever undergone direct selection for very-long-term memory retention of homing routes as professional use of homing pigeons (pigeon-based communications) presumably involved reinforcements on much shorter timescales than years, and recreational uses (pigeon races) usually involve releasing pigeons over unfamiliar areas, where they can not have yet learnt routes (but it is surprisingly difficult to find detailed literature accounts on this, and it would require the work of historians).

With respect to the statistical treatment there needs to be significant changes to how this is presented because as written it is impossible to understand the structure of the models that were fitted, nor assess how appropriate the model fit is - all we have are prose descriptions that are too vague (nobody could reconstruct the models from what is written) and single statistical results without context as to what other parameters or coefficients in the model are saying. There should be either precise descriptions (e.g. 'Route efficiency was modelled as a function of first route task experience (factor with levels experienced/naive) and test route task (factor with levels Greenhill/Horspath) assuming normally distributed errors'), formal equations or R syntax model formulae, and then full model summaries should be presented, and appropriate diagnostics included in the ESM.

We have clarified how we defined our models (l.208-215, l.245-270, l.312-336). Due to the complex structure of our dataset we had to implement quite a large number of different models. Reporting detailed diagnostics for models is not standard practice and would considerably, and in our opinion unnecessarily extend the ESM given the number of (yet simple) models. If you and the editor still believe that this should be included even after we have hopefully clarified both the model definitions and their interpretation, we will comply in future revisions.

I have some concerns that while route efficiency is bounded between 0 and 1, the models appear to be of the normal error family, which has no such constraints on the predictions so I would like to be reassured that the models were not predicting impossible values ('predict' function in R) greater than 1 and less than 0, which would call into question the appropriateness of the models and suggest a different form was needed (for example, beta distributions which are constrained in the interval 0-1).

We added a sentence to clarify that indeed our models do not predict values out of the possible range (l.213-215). Again due to the number of models included it would be laborious to include all diagnostic plots; and we think our interpretations now more accurately align with both the statistics and the biological magnitude of effects which we think is even more important. If you and the editor still would like to see these diagnostics after our present edits, we will comply in future revisions.

Secondly under this topic, the second analysis of 'nearest neighbour' distances (see comments below for discussion of this term), I noted the extremely high degrees of freedom in the test results reported, far exceeding the reported sample sizes (e.g. L295 df=9998). The methods report that this analysis calculated the 'mean distances between pairs of homing routes' but this df value exceeds even the pairwise n you would get if you compared 57 tracks to each other so it is very unclear how this analysis has been conducted to arrive at such a high df. I suspect that it is possible the analysis has been done on individual pairs of points? In either case, the suggestion is either that each track has been entered into the analysis multiple times, or multiple points from each track have been analysed - in either case, this clearly violates the assumption of data independence (using the same track multiple times in a pairwise test, or using multiple points from the same track, which are surely heavily autocorrelated) and suggest to me that this analysis is not valid. The authors need to look again at this, and either describe their analysis with sufficient clarity that a reader can understand how df values of almost 10000 were obtained without violating independence assumptions, or perhaps consult with statisticians about how to model their data such that dependencies are accounted for (e.g. Mantel tests for pairwise comparisons, hierarchical models or GEEs to account for autocorrelation/data dependencies). I suspect that inflated certainty due to lack of accounting for dependence in the data is the reason we see such overlaps in Fig 3 while still obtaining p-values (0.001). I cannot recommend publication until this concern is satisfied since it is fundamental to the author's interpretations.

This was not due to comparison between individual pairs of points but because a large number of releases had been carried in 2016, considerably inflating the number of possible pairs of tracks. We acknowledge that our previous analyses were too imprecise, especially because they biased the representation of pairs of tracks towards those involving solo and pair-control treatments with 60 releases (against 12-24 in other individuals), so not only was there pseudo-inflation but this was different between treatments. We thus now include in our analyses only the last 3 tracks of each individual in 2016 (I.241-244) as well as its unique track in either 2019 or 2020. However we do need to re-use the same tracks in different pairs to investigate "routes" as explained in a previous answer above (between vs within-individual resemblance in paths). This is now a standard practice in e.g. analyses of individual specialization in space use/routes (e.g. Votier et al. 2017 Proc B, <https://doi.org/10.1098/rspb.2017.1068>), in the absence of any better method (and usually authors do not control for the number of tracks per individuals, except in rare exceptions, e.g. <https://doi.org/10.1890/14-1300.1>), the rationale being that the levels of pseudo-replication are comparable between treatments and/or are the very object of inquiry. To further control for the artefactual lowering of p-values we included a complex random structure (I.247-251) and used conservative estimates for degrees of freedom when relevant (I.261, L.269). This random structure still is not perfect but as best as we can given our complex data structure. Indeed this procedure led to much lower but still significant p-values in most cases (I.353-363); moreover as already argued in answers above, the low magnitude of "biological" effect sizes itself warrants careful interpretation, that we hope have been clarified in our new discussion (I.386-409). We hope you will find our biological conclusions to be much better reflections of the data.

Specific comments:

(Please note that we have heavily edited the text so some of these specific comments are no longer directly tractable in the new text but they were accounted for in our edits to help clarify, thanks.)

L33 this study refers to decades, not just 'several years'

Corrected (I.33)

L36 the unity of idea principle suggests this paragraph should be split here when the discussion of memory starts

Corrected (I.37)

L45 specifically, which domains?

Sentence changed

L47 I thought this section of justification was weak - on what basis would the authors a priori expect that there might be differences?

Sentence changed, these justifications are the subjects of the next two paragraphs (I.46-81).

L50 this would be more convincing if there were some methodology presented for the survey? What search terms were used etc the ESM is not clear on this

Corrected: we no longer refer to it as a review (to avoid claims that it would be comprehensive) in the main text, and we added a brief description of the literature search method in ESM. The main point is to show the lack of balance between reports on spatial vs other types of memories over very long time scales.

L58 the authors could consider whether reference to result on humpback migration (humpbacks

<https://royalsocietypublishing.org/doi/full/10.1098/rsbl.2011.0279>

<https://www.frontiersin.org/articles/10.3389/fmars.2020.00414>) are consistent with these

assertions

Thank you for pointing us to these interesting references. Ultimately we chose not to include them as the patterns they report are surprising in terms of the navigational cues that could be used by whales, but are completely silent on whether memory (vs following conspecifics or following a genetic program etc.) is involved, since it does not identify individuals. Our point in this paragraph, that we hope we have clarified (I.48-62), is to expose what we know/don't know about the use of very-long-term spatial memory, rather than the navigational cues.

L61 the quotes around 'true navigation' suggest this terms needs some explanation...

Term deleted, unnecessary

L80 I found this an awkward sentence

Corrected (l.74-75)

L109 naive 'to'?

Reformulated (l.99)

L111-112 as an introduction, this paragraph loses focus here and starts to sound like the methods.. better to end with a clear statement of objectives and predicted outcomes/intepretations...

Condensed (l.82-99)

L118 the level of precision in these positions suggests that the study can measure things to the nearest 10cm, more or less. This seems unlikely - perhaps 3dp would suffice (nearest 100 metre, since the subsequent distance is reported to that level?)

Corrected (l.103)

L153 the sample size for collective learning seems small enough to question the value of including it

As argued in our response to the other reviewer, we think it would simply be a loss of information not to report it, instead of reporting it with full transparency regarding sample size.

L158 do they always fly together? Presumably if birds in these pairs don't then social learning can't happen? Was this assessed?

Yes, we added this precision (l.126-128)

L165 wouldn't 'for the purposes of analysis' be more accurate instead of 'in the Results section' here?

Reformulated

L176 I am confused as to how solo released birds could end up pairing up? It's a bit harsh on the reader to force them to park this thought and wait for pairing up criteria in a later section.... can't this be dealt with all together?

Reformulated (l.180-183); it can unfortunately happen if formerly released pigeons, especially lost ones, circle back over the release point long after vanishing off.

L186 I don't quite get this section - could it be clarified and its rationale/significance explained?

Reformulated (l.292-301, now in Results)

L193 I need some more explanation here as to why this assumption of straight line return following GPS failure is conservative with respect to the analyses reported here?

Details added (l.203-207)

L204 I think 'straight line' would be more cross-culturally accessible here

Corrected throughout

L209 the term 'nearest neighbour distance' is I think pretty suboptimal here - nearly every animal behaviour researchers, and I suspect a lot of others as well, will intuitively read this as relating to a literal nearest neighbour - this is what the term is used for in virtually every other context I have seen it in the literature, meaning the distance to another animal, when as I understand it is nothing of the sort, but instead the distance between a given GPS fix in a focal track to the nearest point on a reference track. Please rethink this terminology

This term has been used extensively in studies quantifying resemblance between paths (where it references the method used to calculate a form of a 'path similarity index'), at least in avian studies. However, we take the Reviewer's point about accessibility on board, and have changed it to Distance Between Paths throughout to be more inclusive of other biologists.

L211 I think we need more clarification of the selection criteria here

Details added (l220-232)

L220 should give model formulation - unclear here how models might be predicted - and identify key predictions/parameters. Also, response is in the range 0-1 - how often did gaussian models predict 'efficiencies' outside this range? I tried to look at the code but found it too poorly commented to properly assess this

This is a valid concern, 100% of predictions were between 0-1 (l.215)

L225 again model descriptions are not detailed enough

Details added

L228 I found this sentence super confusing, I could not understand it - please attempt to clarify

Corrected (I.263-265)

L230 'explored' is too vague here - did you identify the the most similar route? to what end?

Reformulated and moved up (I.149-152)

L254 maybe it is the rendering by the journal but this figure appeared low quality and the text was only just big enough to read, probably would not survive reduction in formatting to column width

Corrected

L254 Also I don't understand why Tueky pariwise tests were needed when a model which had 'Naive' as a baseline in a 'treatment' factor with levels Naive/Indiv/Cult/Coll/Final2016 would not suffice?

We also needed to have Final2016 as a baseline, following your terminology, not just naïve. Plus pairwise comparisons coll vs indiv, indiv vs cult, etc. So we used post-hoc Tukey tests.

L263 it is too unclear specifically what is being tested here for intepretation to really be possible for a naive reader (like me)

Explanations added (I.312-313)

L271 wasn't this already stated above?

Explanations added (I.316-318)

L306 I struggled to understand this sentence please clarify

Corrected (I.373-377)

L317 this is where the eliding of efficiency and route memory really bites - define 'remembered' (seems binary but results suggest gradual decline?)/ What is likely to be the thing(s) that are remembered?

Considerable expansion of discussion on the matter (I.379-409)

L327 could the authors provide any evidence of landmark use from the tracks in this study?

Arguments added (I.386-409, see also answers above)

L328 Really dislike claims of primacy/superiority like this - 'most robust' is vague enough to bring this close to rhetorical anyway. Suggest removing.

Removed throughout

L343 I think the focus on collective learning is excessive here given the sample size

Considerably shortened, more emphasis added on sample size limits (l.410-423)

L403 reference to work by Carroll and colleagues on whale populations struggling to recover migration routes might be pertinent here? <https://www.nature.com/articles/srep16182>

This section of the discussion was considerably simplified and shortened (l.451-453), and we removed most references to comply with the limit on the number of pages.

Appendix B

Thank you for your positive feedback. We have added in the supplementary information a full description of the facilities and procedures for care and husbandry of pigeons (p.16-17). We refer in the main text to this new supplemental section l.103-104; we also added a few words l.171 to explicit that GPS devices weighd less than 5% of pigeon's body mass.